# An Efficient Peptide Screening Method for Mineral-Binding Peptides

**Lam Ian Ku** [1] [ID], **Liza Forbes** [1,2] **and Susana Brito e Abreu** [1,2,*]

1   Julius Kruttschnitt Mineral Research Centre, Sustainable Minerals Institute, The University of Queensland, Indooroopilly, QLD 4068, Australia; lam.ku@uqconnect.edu.au (L.I.K.); l.forbes@uq.edu.au (L.F.)
2   The ARC Centre of Excellence for Enabling Eco-Efficient Beneficiation of Minerals, Shortland, NSW 2307, Australia
*   Correspondence: s.britoeabreu@uq.edu.au

**Abstract:** In mineral processing, arsenic-bearing minerals are particularly difficult to separate from their non-arsenic counterparts because they possess similar surface properties. Peptides are well known for their target specificity and can offer a 'green' alternative to traditional flotation reagents. However, the use of peptide technologies in mineral processing for developing novel flotation reagents has not been explored. Hence, this work aims to develop a screening method to identify mineral-binding peptides as potential reagent candidates. It is hypothesised that peptides can selectively adsorb onto mineral surfaces, and this method can efficiently identify mineral-binding peptides with high specificity toward the target minerals. The methodology presented involves a selection of peptide candidates from existing literature that show affinity toward arsenic species. These peptides were tested for their adsorption performance onto selected mineral surfaces to evaluate their mineral selectivity under flotation conditions. The study demonstrates that the screening method developed is effective in identifying peptides that have an affinity for target minerals, in this case, arsenic minerals. The screening method can be applied to other minerals, thus, unlocking the potential for developing new reagent chemistries for use in mineral processing.

**Keywords:** arsenic minerals; flotation; peptide; collector; flotation reagent; arsenopyrite; pyrite; chalcopyrite; enargite; selective binding





## 1. Introduction

Modern society is heavily reliant on metals that are necessary for vehicles, buildings, machinery, and infrastructure [1]. As a result, the global demand for major metals, such as base metals and precious metals, has increased considerably in recent decades. To maintain the supply of metals, the mining industry needs to process more complex ores as concentrated metal reserves are depleting, and this is often accompanied by an increase in the content of arsenic minerals which are mostly associated with sulphide minerals [2,3]. Arsenic is an undesirable by-product obtained during the metal refinery process. The release of arsenic poses serious health and environmental concerns because of the toxicity and carcinogenicity characteristics of arsenic. Arsenic is listed as the most hazardous substance by the Agency for Toxic Substances and Disease Registry. It is commonly produced as a by-product of treating gold, silver, copper, and other metals [4]. Refineries have restricted emissions to limit the release of arsenic into the environment in the form of effluent gases, fluids, or solids [4,5]. To address this challenge in mineral processing, minimizing the arsenic content during beneficiation processes can significantly reduce the adverse consequences of arsenic emissions to the environment. Therefore, early removal of arsenic-bearing minerals in the processing of ores becomes of paramount importance.

In mineral processing, flotation is one of the most important processes in the concentration of valuable minerals. Finely ground mineral particles enter a flotation cell where they

are mixed with water and chemical reagents. Collectors are one of the most critical reagents used in mineral processing because they modify the surface properties of valuable minerals, making them more hydrophobic and, thus, more likely to attach to air bubbles and float. This enables an increase in selectivity between the desired minerals and the gangue, leading to a higher separation efficiency [6]. The hydrophobic mineral particles will adhere to the air bubbles forming a particle–bubble aggregate that rises to the top of the flotation cell. The floating material containing the valuable minerals is then collected, in a concentrated form [7]. The separation of arsenic-containing minerals from their non-arsenic counterparts is a long-standing challenge in flotation as they possess similar surface properties. Consequently, arsenic minerals have similar flotation behaviour to the valuable minerals with which they are associated [8]. The flotation separation of arsenopyrite (FeAsS) and enargite ($Cu_3AsS_4$) from their non-arsenic counterparts, e.g., pyrite ($FeS_2$) and chalcopyrite ($CuFeS_2$), is of particular interest. Arsenopyrite is the most common arsenic mineral associated with valuable metals, such as gold and copper. Its non-arsenic counterpart, pyrite, is the most abundant sulphide mineral, having significantly lesser value compared to arsenopyrite. Enargite and chalcopyrite both contain a considerable amount of copper. Enargite is the most common copper-containing arsenic sulphide mineral and chalcopyrite is the most abundant copper mineral [9–11].

To improve the separation efficiency of minerals containing arsenic, the key lies in optimizing the selectivity between arsenic-bearing minerals and their non-arsenic counterparts during flotation. Traditional collectors have limitations in achieving this goal, but the target specificity of peptides can potentially offer a solution to circumvent these constraints. A potential solution to this challenge is to develop a highly selective flotation reagent (collector) to target (i.e., adsorb onto) the arsenic-bearing minerals.

In recent years, researchers discovered that particular chains of amino acids can have a high affinity toward certain metals or metal ions [12–15]. Peptides, sometimes referred to as 'mini proteins', are short chains of amino acids linked together in a specific sequence and can comprise up to 50 amino acids. Due to their unique binding specificity, peptides are commonly used in biomedical research for various applications such as drug delivery, drug discovery, biosensors, and cancer treatment. Phage display technology has been used to identify short-binding peptides that exhibit molecular recognition for inorganic materials [16]. Using this approach, researchers have discovered that peptides can have a high affinity toward specific metals and metal ions [12–15]. Some examples include a peptide that targets $MoS_2$ [17], gold-targeting peptides [18–20], a palladium-targeting peptide [21], platinum-targeting peptides [22], silver-targeting peptides [23], titanium-targeting peptides [24,25], metal oxides binding peptides (e.g., iron oxide [26], zinc oxide [27,28]), and mineral binding peptides (e.g., zinc sulphide [29,30] and calcium phosphate [31]). Despite these discoveries, the potential of peptides for the development of mineral processing reagents has yet to be fully explored. For the traditional collectors used in flotation, it is well known that the interactions with the target mineral surfaces involve specific functional groups of the collector molecule, e.g., thiol group, and specific sites of the minerals e.g., $Cu^{2+}$ in chalcopyrite. However, the exact interaction mechanism between the peptide molecules and mineral surfaces is not well understood. Hence, the importance of using a screening method to identify binding peptides.

The target specificity of peptides can offer a potential solution to improve the mineral selectivity and separation efficiency in flotation. Therefore, peptides represent a promising avenue for research in the development of a new generation of flotation reagents. Peptide reagents also offer a 'green' alternative to traditional reagents.

The overall aim of this paper is to present a methodology capable of screening peptides that selectively bind to target minerals, determine the most favourable conditions for the binding and evaluate their potential to recover these minerals in flotation. The screened peptides should then be further tested to quantify their adsorption on the minerals and test their effectiveness under flotation conditions. This is part of our current studies that follow this work.

## 2. Materials and Experimental Methods

### 2.1. Materials

#### 2.1.1. Peptides

The peptides used in this study were synthesised by Mimotopes Pty Ltd. (Mulgrave, Australia) with a purity of ≥95%. All peptides were fluorescent-labelled, containing the 5-carboxyfluorescein (5-FAM) group at the N-terminal and -OH at the C-terminal (5FAM-peptide sequence-OH).

Peptide solutions of 20 μM were prepared by dissolving 1 mg of the lyophilised peptide into a pH-adjusted solvent (milli-Q water or buffer solution) using 0.2 M NaOH to adjust the pH to 5, 7, and 9. The peptide concentration is set to be 20 μM, which has been found to have a maximum fluorescence effect of the 5-FAM fluorescent tag by Tran et al. [32].

#### 2.1.2. Water and Reagents

The resistivity of the Milli-Q water used was 18.2 MΩ·cm with a total organic carbon (TOC) ≤ 3 ppb. For the initial adsorption test screening (as detailed in Section 4.3), the pH of Milli-Q water was adjusted using 0.1 M hydrochloric acid (HCl) and 0.1 M sodium hydroxide (NaOH) to the desired pH value.

Due to observed instability in the behaviour of certain peptide candidates in water, an improvement was made to enhance the experimental repeatability. The buffer solution Britton–Robinson buffer (BRB), which is commonly used in peptide research studies, was introduced. BRB is also known as a "universal" pH buffer which can used for the pH range from 2 to 12. This adjustment aimed to minimise pH fluctuations during adsorption tests, thereby promoting a more controlled and stable environment for peptide–mineral binding.

The buffer solution was prepared with 0.04 M boric acid ($H_3BO_3$), 0.04 M phosphoric acid ($H_3PO_4$), and 0.04 M acetic acid ($CH_3COOH$) and adjusted to the desired pH (pH 5, 7 and 9) with 0.2 M sodium hydroxide (NaOH).

Methyl isobutyl carbinol (MIBC), a common flotation frother, was added to the peptide solution in the third round of screening tests. The MIBC concentration was 10 μL MIBC/L peptide solution, which mimics the frother dosage in mineral flotation.

#### 2.1.3. Minerals

Arsenopyrite, pyrite and chalcopyrite were purchased from Geo Discoveries (West Gosford, NSW, Australia) with a purity of ≥98%. Enargite was sourced from the geological collection from The Commonwealth Scientific and Industrial Research Organisation (CSIRO) with a purity of ≥67%, based on the mineral liberation analysis (MLA) and assay analyses. The flat surfaces of the minerals were prepared by moulding them in a block of resin and cutting them into slices using a diamond saw. The mineral slices have a diameter of approximately 3 cm and a thickness of approximately 4 mm.

### 2.2. Techniques

#### 2.2.1. Fluorescence Microscopy

Fluorescence microscopy was used in this study to examine the presence of peptide adsorbed onto the mineral surfaces. The fluorescence microscope used was a Nikon Eclipse TS-100 (Tokyo, Japan) inverted microscope with a Tucsen H series microscope camera (Fuzhou, China).

As the excitation and emission wavelength of the fluorescein (5-FAM) were determined to be 492 nm and 517 nm, respectively [32], the green excitation filter of the microscope was selected. The software TCapture (version 4.3.0.605) was used for the acquisition and processing of fluorescence images.

#### 2.2.2. pH and Eh Measurements

The pH and redox potential (Eh) of the peptide solutions were monitored during the adsorption tests. The probes used for the pH and Eh measurements were a TPS Pt ORP

Electrode probe and a pH probe connected to a TPS 90-FLMW Field Lab Analyzer meter (TPS, Brendale, QLD, Australia).

### 2.2.3. Optical Tensiometer

The optical tensiometer (Kruss Drop Shape Analyzer DSE10, Hamburg, Germany) was used to quantify the hydrophobicity of the mineral surfaces, with and without adsorbed peptide. To measure the contact angle, the sessile drop method was used. In the measurement process, deionised water droplets were placed on the mineral surface and the contact angles were determined by the instrument software automatically. More than eight spots were measured on each mineral surface and the average contact angle was calculated.

### 2.2.4. Image Processing Software

The quantification of peptide coverage on the mineral surface can be calculated using the images obtained by the fluorescent microscope and processed using the imaging software, Image J. The "threshold" parameter that identifies the fluorescent area in Image J is set as 50 for all the fluorescent images.

## 3. Method Development

In this section, a novel approach for screening peptides that successfully bind to mineral surfaces is presented. This method first identifies the possible binding species on the target mineral surface. From the literature, appropriate peptide sequences with a binding affinity toward arsenic species will be selected and synthesised with a fluorescence tag. The adsorption performance and mineral selectivity of these peptides will be tested on flat mineral surfaces of selected minerals. The successful peptides will be considered potential flotation reagent candidates and will undergo further characterisation studies. A detailed description of each stage of the method is provided in the following sections.

### 3.1. Identification of the Target Binding Species on the Mineral Surface

The first step of the screening process is to identify the surface species that are expected to be present on the mineral of interest and have the potential to differentiate it from other minerals to achieve selectivity. In this work, we aim to distinguish between arsenopyrite and pyrite, as well as enargite and chalcopyrite. Therefore, the target surface species under solution conditions are the various arsenic ion species. Arsenic is a metalloid that can be found in the oxidation states of +V (arsenate), +III (arsenite), 0 (arsenic), and −III (arsine) [33]. The most important factors controlling arsenic speciation in an aqueous solution are the redox potential (Eh) and pH [34]. In aqueous form, the dominant arsenic oxyanions are [35]:

- Arsenite, under reducing conditions, which includes $H_3AsO_3$, $H_2AsO_3^-$, $HAsO_3^{2-}$ and $AsO_4^{3-}$ ions;
- Arsenate, under oxidizing conditions, which includes $H_3AsO_4$, $H_2AsO_4^-$, $HAsO_4^-$, $HAsO_4^{2-}$, $HAsO_4^{2-}$ and $AsO_4^{3-}$ ions.

Hence, a literature review is conducted to identify peptide sequences that bind to any form of the above species.

### 3.2. Identification of Peptide Sequences That Bind to the Target Species

Following an extensive review of the literature across various fields of peptide applications, several peptide sequences were identified to have an affinity toward arsenic species and enargite (arsenic mineral). The list of peptides selected from the literature is shown in Table 1.

**Table 1.** Summary of the selected arsenic-binding peptides based on the literature.

| Category | Target Substance (i.e., Selective to) | Amino Acid Sequence | The Identified Peptide Can Distinguish the Target Substance from | Reference | Peptide Candidate Code * |
|---|---|---|---|---|---|
| Enargite binding peptides | Enargite (Cu$_3$AsS$_4$) | MHKPTVHIKGPT | Silica and Chalcopyrite | [13] | PEng-1 |
| | | NPEHAAFSPVTV | Pyrite and Chalcopyrite | Patent WO/2018/052134 | PEng-2 |
| | | SKDGAGAAKRTS | Pyrite and Chalcopyrite | Patent WO/2020/085219A1 | PEng-3 |
| Arsenic-ion binding peptides | As(III) | TQSYKHG | As(V), Zn$^2$, Cd$^{2+}$, Fe$^{3+}$ and Cu$^{2+}$ | [36] | PAs3 |
| | As(III) and As(V) | TPSGDMQ | Zn$^2$, Cd$^{2+}$, Fe$^{3+}$ and Cu$^2$ | | PAs35 |
| Arsenic-oxyanions binding peptides | Arsenic oxyanions | FHMPLTDPGQVQ | No information could be found. | [37] | PAsO-1 |
| | | SIHSVTKGRYPV | | | PAsO-2 |

*: Candidate code format "PXXX-A"; 'P' is the short form of peptide identified by phage display; 'XXX' represents the target speciation (Eng = Enargite, As3 = As(III), As35 = As(III) and As(V), AsO = arsenic oxyanions); 'A' is the candidate number within the same target speciation, e.g., PAsO-1 vs PAsO-2.

The selected peptides are categorised into three types according to the target speciation:

- The enargite binding peptides: PEng-1, PEng-2, PEng-3;
- The arsenic-ion binding peptides: PAs-3, PAs-35;
- The arsenic-oxyanion binding peptides: PAsO-1, PAsO-2.

### 3.3. Synthesis of the Peptide Sequences with a Fluorescent Tag

All the peptide candidates listed in Table 1 were synthesised by Mimotopes Pty Ltd. The peptides were labelled with 5-Carboxyflorescein (5-FAM) at the N-terminus. The fluorescent tag was added to enable the visualisation of the peptides on the mineral surface using a fluorescence microscope. These peptides were used to conduct the peptide-mineral adsorption studies described in the next stage.

### 3.4. Testing of the Peptide Adsorption on Flat Mineral Surfaces

The main objective of this stage is to study peptide–mineral binding under varying conditions. To achieve this objective, a series of adsorption studies were conducted on flat mineral surfaces of arsenopyrite, pyrite, chalcopyrite, and enargite. The protocol established for the adsorption test provides a rapid screening of the peptides to identify successful binding onto the mineral surfaces. It is used to investigate the effect of variables such as pH and the presence of chemical reagents or solution species on the binding performance.

The procedure for peptide adsorption is illustrated in Figure 1 and described in detail below.

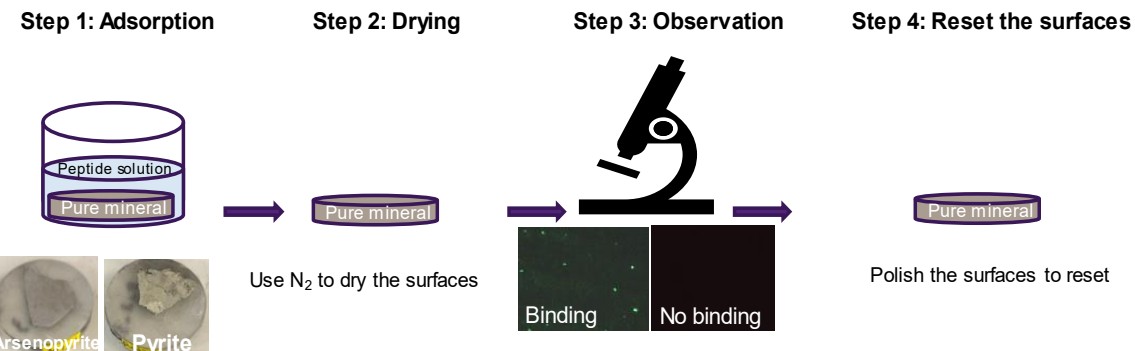

**Figure 1.** Schematic of the peptide adsorption test procedure.

To assess the peptide–mineral interactions under different conditions, including conditions that closely relate to mineral flotation (i.e., presence of frother in solution), and determine the optimal binding conditions, the following variables were tested in this work:

- Two peptide solvents: Milli-Q water and Britton-Robinson buffer solution (BRB);
- Three pH conditions: pH 5, 7 and 9;
- Four minerals: arsenopyrite, pyrite, enargite and chalcopyrite;
- Presence of methyl isobutyl carbinol (MIBC) in solution, the most common flotation frother (10 µL MIBC/L peptide solution).

The main steps involved in this procedure are described below:

- Step 1: Adsorption of peptide

Prior to the peptide adsorption tests, the pure mineral surfaces were visualised under the fluorescence microscope to identify if any natural fluorescence was present before performing the peptide adsorption studies. No natural fluorescence was observed for the minerals tested, as the images appeared completely black.

For the adsorption tests, the clean, flat mineral surfaces were immersed in the peptide solution for 5 min.

- Step 2: Drying of surfaces

The surfaces were then dried with nitrogen gas to remove the residual solution from the surface. The surfaces were then rinsed with Milli-Q water to remove any physically adsorbed peptide (not bound to the surfaces) and dried under nitrogen gas again.

- Step 3: Observation under the microscope

The dried mineral slices were placed onto the sample holder of the fluorescence microscope. The peptide present on the mineral surface can be observed as green, fluorescent spots under the fluorescence microscope.

- Step 4: Surface Resetting

Resetting of the mineral surfaces was performed between tests to ensure no reagent residue was left on the surfaces from the previous experiments. Two resetting methods were tested to determine the most suitable method:

(a) Soaking the mineral surface containing the adsorbed peptide in the following solutions: Milli-Q water, 90% ethanol, 100% ethanol, acetone, 0.5 M/1 M/2 M sodium hydroxide, 0.1 M/1 M hydrochloric acid, 0.5 wt.% Tween 80 and glycine elution buffer (0.2 M Glycine-HCl at pH 2.2 with 1 mg/mL bovine serum albumin);

(b) Polishing the mineral surfaces using a 3-micron cloth polishing disc, 80-grit and 180-grit sand polishing discs. After the polishing, the mineral surfaces were rinsed with Milli-Q water, dried with nitrogen gas and observed under the fluorescence microscope to detect the presence of any residue of the peptide.

Among the methods tested above, the best surface resetting results were achieved by polishing the mineral surfaces with a 180-grit polishing disc. Hence, this was the method used for all adsorption tests performed.

- Step 5: Selection of peptide candidates as potential flotation collectors

After confirming the peptide adsorption and semi-quantifying the successful binding on the mineral surfaces, the peptide candidate was selected or rejected for the next stage based on the adsorption outcome and mineral selectivity.

One of the most important properties of flotation collectors is the ability to increase the hydrophobicity of the target mineral. Hence, it is crucial to determine whether the peptide will increase or decrease the natural mineral hydrophobicity upon adsorption. Therefore, we need to quantify the surface hydrophobicity before and after the peptide adsorption.

## 4. Results

### 4.1. Testing Conditions

The peptide–mineral interactions were studied through a series of adsorption tests conducted under different testing conditions. The first screening round included the seven peptide candidates selected from the literature. The adsorption performance of these peptides was tested using pure arsenopyrite and pyrite surfaces, arsenopyrite being the target mineral. The peptides that showed successful binding in this round were further tested in the second round of the adsorption test. In the second round, an additional pair of arsenic/non-arsenic minerals was introduced to further evaluate the binding selectivity. The buffer solution was used to stabilise the pH of the peptide solutions. The third round of the adsorption tests introduced the presence of MIBC in the peptide solution, which is a common frother used in flotation.

A summary of the peptide screening procedure is presented in Figure 2.

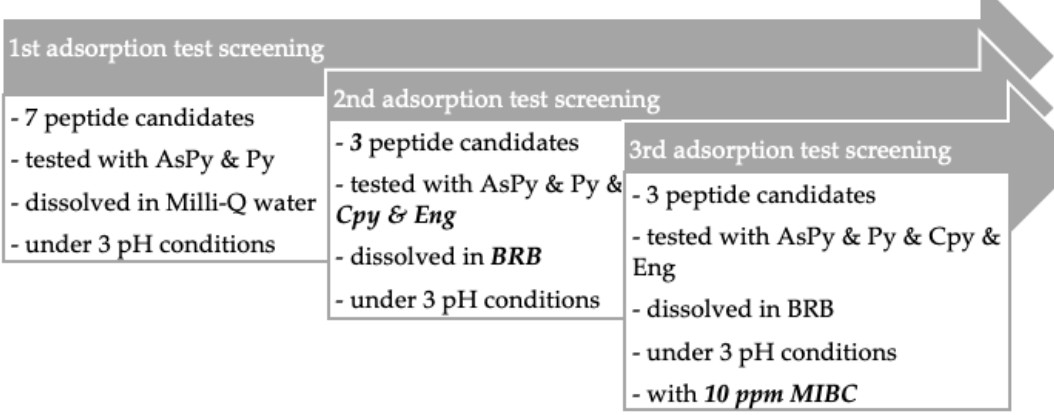

**1st adsorption test screening**
- 7 peptide candidates
- tested with AsPy & Py
- dissolved in Milli-Q water
- under 3 pH conditions

**2nd adsorption test screening**
- *3 peptide candidates*
- tested with AsPy & Py & *Cpy & Eng*
- dissolved in *BRB*
- under 3 pH conditions

**3rd adsorption test screening**
- 3 peptide candidates
- tested with AsPy & Py & Cpy & Eng
- dissolved in BRB
- under 3 pH conditions
- with *10 ppm MIBC*

AsPy: arsenopyrite; Py: pyrite; Cpy: chalcopyrite; Eng: enargite

*Parameters in Italics*: the new variables that were introduced into the new round of the adsorption test

**Figure 2.** Summary of the adsorption test screening procedure.

### 4.2. Semi-Quantification of Peptide Adsorption

Since the peptides were fluorescence-labelled, the fluorescence observed on the mineral surfaces after each adsorption test indicates positive binding of the peptide. The relative amount of fluorescence observed on the mineral surfaces is used as a semi-quantification of the peptide adsorbed. As an example of the semi-quantification observed under the fluorescence microscope, selected images of the fluorescence detected on the surfaces are presented in Figure 3.



**Figure 3.** Semi-quantification of the peptide adsorption was observed as indicated by the amount of fluorescence detected on the mineral surfaces. (**a**) no fluorescence observed; (**b**) observed a small amount of fluorescence; (**c**) observed a moderate amount of fluorescence; and (**d**) observed a high amount of fluorescence. The images are examples taken from various test conditions.

In this study, we consider three levels of peptide observed on the surfaces: a small amount, a moderate amount, and a high amount of fluorescence. As such, the presence of peptides on the surfaces was categorised into:

a.　　No fluorescence observed: ×
b.　　A small amount of fluorescence is observed: √
c.　　A moderate amount of fluorescence is observed: √√
d.　　A high amount of fluorescence is observed: √√√

*4.3. The 1st Adsorption Test Screening—Peptide Dissolved in Milli-Q Water*

In the first round of adsorption test screening, the seven peptide candidates listed in Table 1 were tested for their adsorption performance on the mineral surfaces of arsenopyrite and pyrite and the results are shown in Table 2. To ensure the reproducibility of the results, two to four independent tests were conducted to evaluate each adsorption condition. In some cases, fluctuations in the pH were observed during the adsorption tests, which explains some variation of the adsorption observed between tests, e.g., as observed with PAsO-1. To minimise this effect, a buffer solution was introduced in the second round of the adsorption tests.

**Table 2.** Summary of the 1st adsorption test screening results.

| PEPTIDE | ARSENOPYRITE | | | PYRITE | | |
|---|---|---|---|---|---|---|
| | pH 5 | pH 7 | pH 9 | pH 5 | pH 7 | pH 9 |
| PENG-1 | ×/√/√√/√√ | ×/√/√√√/√√√ | ×/√/√/√ | √√/√/√/√√ | √/√√/√/× | √√/×/× |
| PENG-2 | ×/× | ×/× | ×/× | ×/× | ×/× | ×/× |
| PENG-3 | ×/× | ×/× | ×/× | ×/× | ×/× | ×/× |
| PASO-1 | ×/×/×/√ | √/×/×/× | ×/√/×/× | ×/×/×/× | ×/×/×/× | ×/×/×/× |
| PASO-2 | √√√√/√√√/√√ | √√/√/√/√ | √√/×/× | √√/√√/√/√√ | √√/√/√ | √√/×/× |
| PAS3 | ×/× | ×/× | ×/× | ×/× | ×/× | ×/× |
| PAS35 | ×/× | ×/× | ×/× | ×/× | ×/× | ×/× |

×: No fluorescence was observed; √: A small amount of fluorescence is observed; √√: A moderate amount of fluorescence is observed; and √√√: A high amount of fluorescence is observed.

From the above results, it can be observed that from the seven peptides tested, only Peng-1, PasO-1, and PAsO-2 show positive binding onto both arsenopyrite and pyrite surfaces under the pH range tested. The adsorption of PEng-1 and PAsO-2 was found to be considerably more pronounced than PAsO-1.

The adsorption selectivity against arsenopyrite was observed for the peptides PEng-1 at pH 7 and 9, and PAsO-2 at pH 5. PAsO-1 shows weak but specific binding only to arsenopyrite surfaces. The binding appeared at pH 5, 7, and 9. Therefore, PEng-1, PAsO-1, and PAsO-2 show selective binding and will proceed to the next phase of the screening in the presence of the buffer solution BRB.

In terms of the peptide-targeted ion species, no binding was observed for the arsenic-ion binding group, indicating that no (significant) amount of $As^{3+}$ and/or $As^{5+}$ ions were present on the mineral surfaces under the tested conditions. Interestingly, both PAsO-1 and PAsO-2, which are both arsenic oxyanion binding peptides, show specific binding onto arsenopyrite. This agrees with the arsenopyrite surface speciation expected under the conditions tested. Arsenate is expected to form on top of an iron hydroxide layer on arsenopyrite surfaces, as reported in previous studies [38–45].

The pH-Eh diagrams of arsenopyrite and pyrite in aqueous solutions were used to plot the successful binding conditions (Figure 4). Figure 4 denotes successful bindings: stars represent PEng-1, triangles denote PAsO-1, and circles with a cross indicate PAsO-2 bound to the minerals. The colours—black for pH 5, white for pH 7, and grey for pH 9—reflect the respective pH conditions. It can be observed that most of the data points fall into the "Ferrihydrite, $Fe(OH)^{-3}$, Fe arsenates, +As-ferrihydrite" area for arsenopyrite (in Figure 4 left), where ferrihydrite can be $5Fe_2O_3 \cdot 9H_2O$ or generally represent a hydrated form of iron oxide/hydroxide, and "$Fe(OH)_3$" for pyrite (in Figure 4 right). Based on the successful binding of both arsenite/arsenate-targeting peptides in the experiments, it can be concluded that arsenite/arsenate ion species were the key binding species and

these are likely to be formed on top of an iron hydroxide layer under the experimental conditions. However, one should note that peptide candidates that did not show binding under the tested conditions may still be able to bind to arsenopyrite under different pH and Eh conditions outside of the tested range.

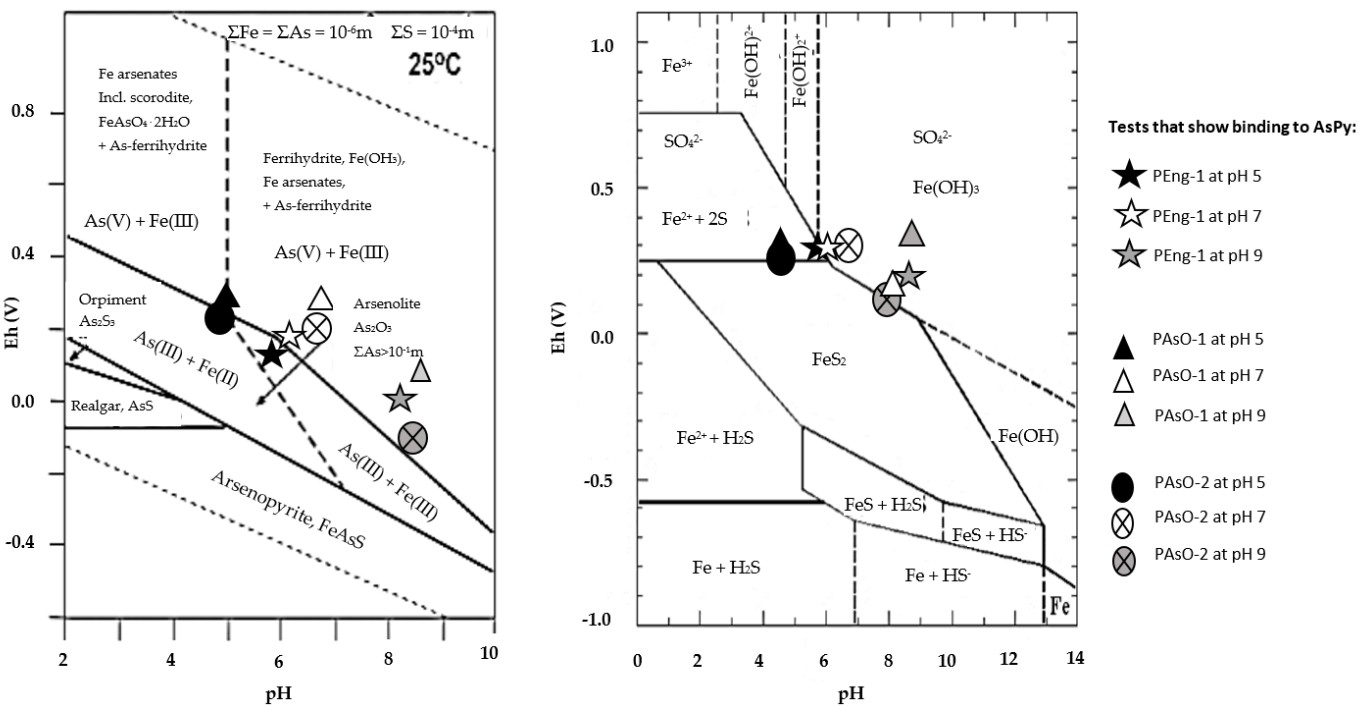

**Figure 4.** The binding conditions observed on the arsenopyrite (**left**) and pyrite (**right**) mineral surfaces are plotted on the respective pH−Eh diagrams (S.H.E.) at 25 °C. The diagrams were reproduced from Craw (2003) [46] and Moslemi and Gharabaghi (2016) [47].

### 4.4. The 2nd Adsorption Test Screening—Peptide Dissolved in pH-Adjusted BRB

The three peptides that showed positive binding to arsenopyrite, PEng-1, PAsO-1, and PAsO-2, were re-tested in the presence of the buffer solution BRB to better control the pH during the experiments. This round of adsorption tests aims to evaluate if BRB increases the stability of the peptides in solution by minimising pH fluctuations, thus increasing the experimental repeatability. This adsorption round will also study the peptide binding using an additional pair of arsenic/non-arsenic minerals, enargite, and chalcopyrite to further test mineral selectivity.

In this set of experiments, the pH of the BRB solution was adjusted to 5, 7, and 9 and used to dissolve the peptides. The adsorption test results are summarised in Table 3. To ensure the reproducibility of the results, two to three independent tests were conducted to evaluate each adsorption condition.

**Table 3.** Summary of the adsorption test results with peptides dissolved in buffer solution (BRB).

| | PAsO-1 with BRB | | |
|:---:|:---:|:---:|:---:|
| **Minerals** | **pH 5** | **pH 7** | **pH 9** |
| AsPy | √√√/√√√√/√√ | √/×/√ | √/×/√ |
| Py | √/√/√ | √/√/√ | ×/×/× |
| Eng | √/√√/√ | ×/×/× | ×/×/× |
| Cpy | √√/√√/√√ | ×/×/× | ×/×/× |

**Table 3.** *Cont.*

| | PEng-1 with BRB | | |
|---|---|---|---|
| | **pH 5** | **pH 7** | **pH 9** |
| AsPy | √√√//√√√//√√√ | √√//√√//√ | √√//√//√√ |
| Py | √//√//√ | √//√//√ | √√//√//√√ |
| Eng | ×/× | ×/× | ×/× |
| Cpy | √√//√√//√√ | √//√//√√ | √√//√√//√√ |
| | PAsO-2 with BRB | | |
| | **pH 5** | **pH 7** | **pH 9** |
| AsPy | √√√//√√//√√ | √//×/√ | √//√//× |
| Py | √//√//√ | √//√//√ | ×/√//√ |
| Eng | √//√√//√ | ×/√//√ | ×/√//× |
| Cpy | √//√//√ | √//√//√ | √//√//× |

×: No fluorescence was observed. √: A small amount of fluorescence is observed. √√: A moderate amount of fluorescence is observed. √√√: A high amount of fluorescence is observed. AsPy: arsenopyrite; Py: pyrite; Eng: enargite; Cpy: chalcopyrite.

From the results presented in Table 3, the adsorption performance of the peptides is found to be more reproducible with the use of BRB, as consistent results were obtained with the repeat tests. PEng-1 was found to be more selective toward arsenopyrite at pH 5 and did not adsorb onto enargite at any pH value. For PAsO-1, the selectivity toward arsenopyrite can be observed at pH 5 and 7; PAsO-1 did not bind to either enargite or chalcopyrite at pH 7 and 9. PAsO-2 is relatively unselective as it adsorbed onto the four minerals under all conditions, except for enargite at pH 9.

*4.5. The 3rd Adsorption Test Screening—Peptide Dissolved in pH-Adjusted BRB with Frother MIBC*

This set of experiments was repeated in the presence of 10 ppm of Methyl Isobutyl Carbinol (MIBC), a commonly used frother in mineral flotation. The aim of this round of tests is to assess if there is any impact of the presence of MIBC on the peptide–mineral interactions. The adsorption test results are presented in Table 4.

**Table 4.** Summary of the adsorption test results with peptide dissolved in buffer solution (BRB) in the presence of MIBC.

| | PEng-1 with BRB and MIBC | | |
|---|---|---|---|
| **Minerals** | **pH 5** | **pH 7** | **pH 9** |
| AsPy | √√//√√//√√√ | √√//√√√//√√ | √//√//√ |
| Py | √//√//√√ | √//√//√ | ×/√//√ |
| Eng | ×/× | ×/× | ×/× |
| Cpy | √√//√√//√√ | √√//√√//√√ | √√//√//√ |
| | PAsO-1 with BRB and MIBC | | |
| | **pH 5** | **pH 7** | **pH 9** |
| AsPy | √√√//√√√//√√ | √//×/√ | ×/×/× |
| Py | √√//√√//√√ | ×/√//√ | ×/×/× |
| Eng | √//√//√ | ×/×/× | ×/×/× |
| Cpy | √√//√√//√√ | ×/×/× | ×/×/× |

**Table 4.** *Cont.*

| | PAsO-2 with BRB and MIBC | | |
| --- | --- | --- | --- |
| | **pH 5** | **pH 7** | **pH 9** |
| AsPy | √/√√/√√ | √/×/√ | √/√/× |
| Py | √√/√/√√ | √/√√/√ | ×/√/√ |
| Eng | √/×/√ | √/×/× | ×/×/× |
| Cpy | √/√/√ | √/√/√ | √/√/√ |

×: No fluorescence was observed. √: A small amount of fluorescence is observed. √√: A moderate amount of fluorescence is observed. √√√: A high amount of fluorescence is observed. AsPy: arsenopyrite; Py: pyrite; Eng: enargite; Cpy: chalcopyrite.

Based on the results of the adsorption tests, it can be concluded that MIBC does not have an adverse impact on peptide selectivity as the adsorption performance was not affected.

Figure 5 shows selected images obtained from the fluorescence microscope for the tests performed above. The results presented in Table 4 were obtained at pH 5 with PEng-1 in BRB, where images (a) to (d) were obtained in the absence of frother and images (e) to (h).

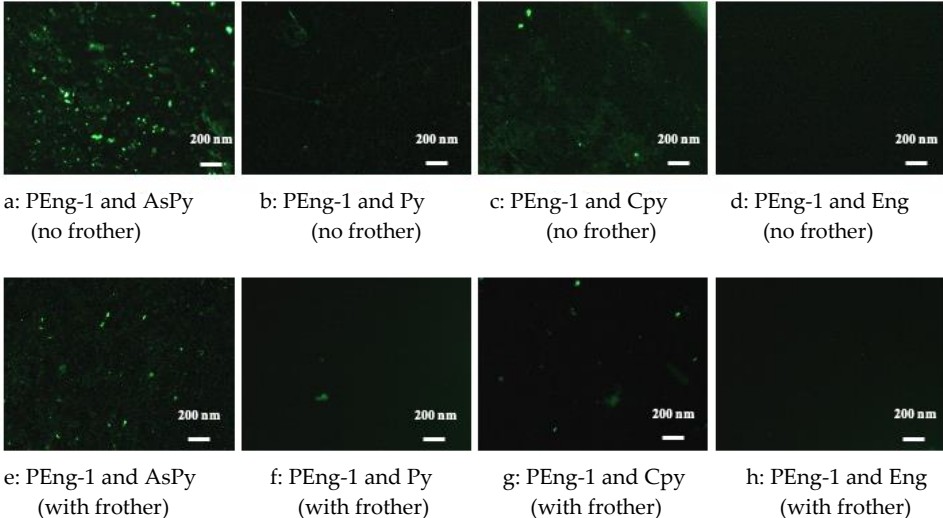

a: PEng-1 and AsPy (no frother)   b: PEng-1 and Py (no frother)   c: PEng-1 and Cpy (no frother)   d: PEng-1 and Eng (no frother)

e: PEng-1 and AsPy (with frother)   f: PEng-1 and Py (with frother)   g: PEng-1 and Cpy (with frother)   h: PEng-1 and Eng (with frother)

**Figure 5.** Adsorption results observed from the fluorescent microscope in the presence and absence of frother, MIBC (objective lens: 4×). The scale bars represent 200 nm. (**a**). arsenopyrite surface after the adsorption study using PEng-1. (**b**). pyrite surface after the adsorption study using PEng-1. (**c**). chalcopyrite surface after the adsorption study using PEng-1. (**d**). enargite surface after the adsorption study using PEng-1. (**e**). arsenopyrite surface after the adsorption study using PEng-1 in the presence of MIBC. (**f**). pyrite surface after the adsorption study using PEng-1 in the presence of MIBC. (**g**). chalcopyrite surface after the adsorption study using PEng-1 in the presence of MIBC. (**h**). enargite surface after the adsorption study using PEng-1 in the presence of MIBC.

The results of the screening method enabled selection of promising peptide candidates from the initial list of peptides tested, i.e., the peptides that are selectively bound to the target mineral arsenopyrite. PEng-1 peptide was identified as the best candidate that shows selective adsorption toward arsenopyrite and, to a lesser extent, chalcopyrite.

*4.6. Quantification of Peptide Coverage*

While the rapid screening method can effectively identify successful binding peptides, in order to undertake a more quantitative comparison of peptide adsorption, the image processing software, Image J, can be used to quantify the peptide coverage on the surface as the proportion of fluorescent surface area determined from the fluorescent microscope images.

As an example, the results in Figure 6 present the peptide coverage area on arsenopyrite (a) and pyrite (b) after adsorption with PEng-1 in buffer solution at pH 5. From the results calculated by Image J, the peptide coverage area of PEng-1 on arsenopyrite was 2.26% and the peptide coverage area of PEng-1 on pyrite was 0.27%. This result further confirms the selectivity of PEng-1 toward arsenopyrite under the tested conditions.

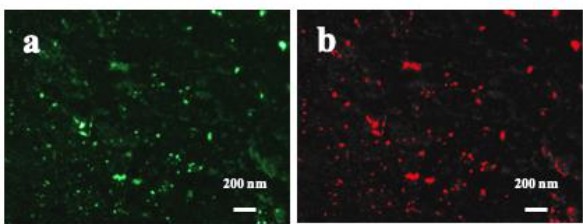

a: Original fluorescence image (Aspy)  b: Binding area (in red) detected by Image J
(Aspy with PEng-1 in buffer solution at pH 5)
Coverage Area% measured by Image J = 2.259%

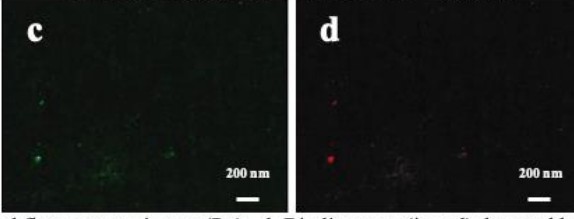

c: Original fluorescence image (Py)  d: Binding area (in red) detected by Image J
(Py with PEng-1 in buffer solution at pH 5)
Coverage Area% measured by Image J = 0.270%

**Figure 6.** Peptide binding area and coverage percentages determined by the image processing software, Image J. (**a**): Original fluorescence image from the adsorption test of arsenopyrite with PEng-1 in buffer solution at pH 5. (**b**): Binding area (in red) of Figure 6a detected by Image J. (**c**): Original fluorescence image from the adsorption test of pyrite with PEng-1 in buffer solution at pH 5. (**d**): Binding area (in red) of Figure 6c detected by Image J.

From all the results presented above, PEng-1 is found to be the best candidate to adsorb selectivity onto arsenopyrite. This peptide will be further studied in the next section to quantify the hydrophobicity of the peptide-adsorbed mineral surfaces.

### 4.7. Contact Angle Measurement

The contact angle measurements of the flat mineral surfaces were conducted after the PEng-1 peptide adsorption under conditions that showed positive binding.

The contact angle measurements conducted in this study are presented in Figure 7. More than eight spots were measured on each mineral surface and the average contact angle was calculated. All the mineral baseline contact angles measured, i.e., the natural mineral hydrophobicity, agree with the expected values from the literature [48–54].

In Figure 7a, it can be observed that the contact angle of arsenopyrite increased after the adsorption tests, from $76.9 \pm 6.4°$ to $85.4 \pm 3.5°$. The statistical two-sample t-test resulted in a $p$-value of 0.02, indicating that the hydrophobicity of arsenopyrite after adsorption with PEng-1 is statistically different from the baseline.

In Figure 7b, the contact angle of pyrite remains the same after adsorption, being $82.9 \pm 8.3°$ before and $82.5 \pm 3.8°$ after adsorption. The $p$-value of 0.46 confirms there is no statistical difference between the contact angle values.

In Figure 7c, the contact angle measurement rises slightly on the chalcopyrite surface, from $83.0 \pm 2.5°$ to $85.0 \pm 2.8°$. The $p$-value of 0.15 indicates poor evidence that the hydrophobicity of the peptide-adsorbed chalcopyrite is different from the baseline.

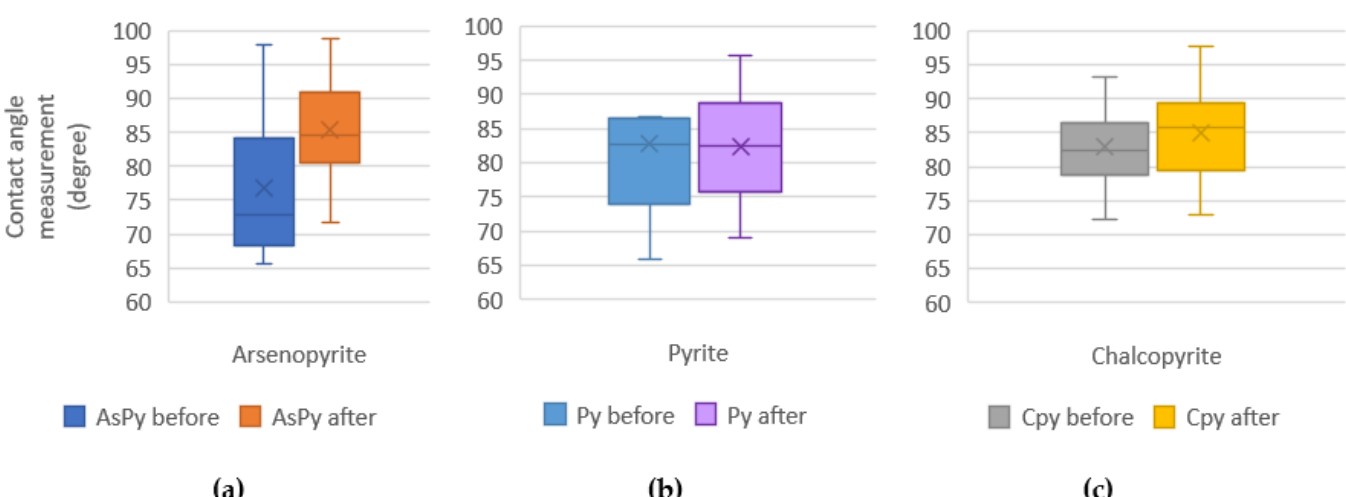

**Figure 7.** Contact angle values of the mineral surfaces before and after PEng-1 adsorption tests: (**a**) arsenopyrite; (**b**) chalcopyrite (**c**) pyrite. This experiment was performed in triplicate, the data and error bars are the average values and the standard deviation (within $+/-$ 5% of the average values).

The contact angle results agree with the screening tests which showed higher adsorption of PEng-1 onto arsenopyrite, and chalcopyrite to a lesser extent. The results also indicate that the hydrophobicity of the peptide PEng-1 is likely around 85° but since the natural hydrophobicity of the fresh mineral surfaces is also high under the pH tested, the peptide may not increase the mineral hydrophobicity sufficiently to produce differences in the minerals floatability. This difference could be improved by modifying the peptide structure to increase its hydrophobicity or reducing the natural hydrophobicity of the minerals (e.g., by pre-oxidation). Such modifications would form part of the next stage in the peptide development as a flotation collector.

*4.8. Discussion*

The aim of this work was to develop a rapid screening method to identify peptides that bind to the target mineral surfaces, in this case, arsenic-bearing sulphide minerals. This investigation was guided by the following objectives:

1.  Identify peptide sequences that bind selectively to the target minerals and the pH conditions to achieve optimal binding, i.e., peptide adsorption on the mineral surfaces.
2.  Test the peptide binding performance under flotation conditions (i.e., the presence of frother).
3.  Assess the hydrophobicity properties of the peptide.
4.  Establish proof-of-concept for the method developed to select peptide candidates as potential flotation reagents to selectively recover the minerals of interest.

In this study, a novel and efficient method for screening and identifying mineral-binding peptides was developed. The method consists of five stages, as summarised in Figure 8.

The method takes advantage of the binding specificity of peptides to selectively target the minerals of interest, in this case, arsenic minerals. It was hypothesised that peptides bind selectively to specific chemical species, which are the arsenic species in this study. The results presented show that from the seven peptides initially selected for screening, PEng-1 was found to selectively bind to the target mineral, arsenopyrite, and was identified as a promising reagent candidate for targeting arsenopyrite in flotation.

This study also showed that peptide sequences could bind to mineral surfaces with different degrees and that the adsorption performance is highly driven by the conditions tested, especially pH. The results demonstrate the effectiveness of the screening method in identifying successful peptide binding onto flat mineral surfaces under varied conditions. Moreover, the adsorption of peptides can be quantified using image processing software

such as Image J to provide the peptide coverage on the surface. This work also quantified the mineral hydrophobicity changes that resulted from the peptide adsorption on the surface. It was found that the peptide binding to arsenopyrite, PEng-1, was more hydrophobic than the mineral, thus, representing a promising candidate as a flotation collector for arsenopyrite.

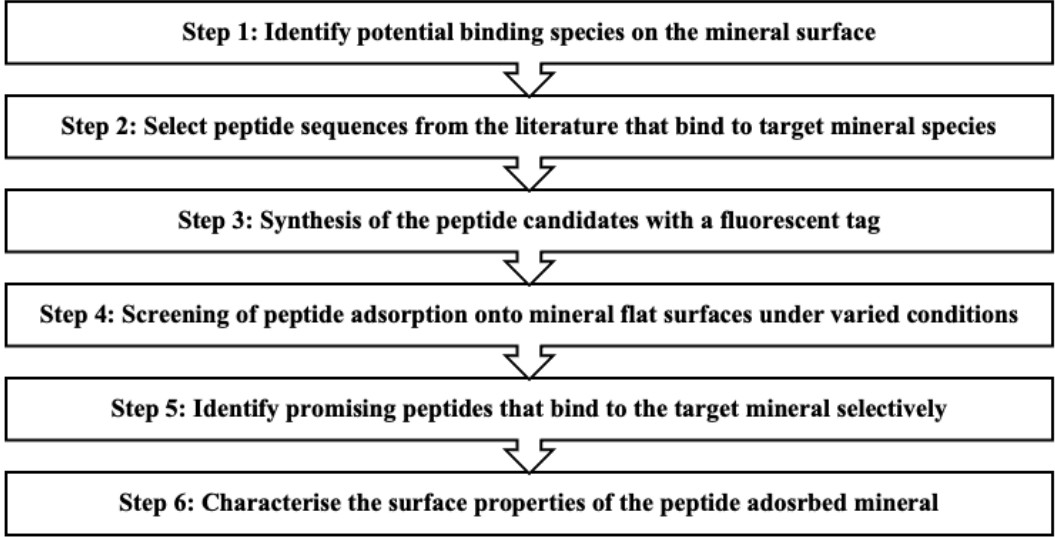

**Figure 8.** Summary of the peptide screening method.

The introduction of buffer solutions markedly enhanced the stability of peptides in the solution mitigating pH fluctuations and improved the repeatability of the adsorption experiments. Notably, the binding and selectivity of peptides were not affected by the presence of the frother MIBC in the solution. This suggests that peptide-mineral interactions are stable under flotation conditions, and therefore, can be further tested in mineral flotation.

The findings above support the method presented as an efficient peptide screening for identifying peptides with specific binding for target minerals.

*4.9. Future Work*

This study represents an initial foray into the application of peptide technologies within the domain of mineral processing. Successfully identifying peptides exhibiting promising affinity for arsenic-bearing minerals through this screening method marks the commencement of a new area of research to investigate the application of peptides as flotation reagents.

The subsequent phase of this work involves a comprehensive characterization of peptide–mineral interactions and adsorption onto mineral surfaces. This in-depth surface characterisation analysis can then provide insights into the peptide–mineral binding mechanism such as identifying the key binding amino acids in the peptide sequence.

Furthermore, forthcoming investigations following this work will focus on quantifying the adsorption process to identify the minimum peptide concentration needed for optimal selectivity and testing the peptides in practical flotation conditions.

In essence, while this paper serves as a critical starting point for the investigation of peptides as flotation reagents, the exploration into peptide-based reagents for mineral processing remains an ongoing pursuit. Subsequent phases of this research endeavour aim to deepen the comprehension of peptide–mineral interactions and pave the way for the development of highly efficient and selective flotation reagents, propelling the field towards more sustainable and environmentally conscientious practices.

**5. Conclusions and Recommendations**

The method developed in this paper has been demonstrated to successfully identify peptide sequences that bind to target minerals, in this case, arsenopyrite, and the most

favourable conditions for optimal binding. The 'screened' peptide PEng-1 represent a promising candidate to be investigated as a flotation reagent, with mineral collecting properties, in this case.

Compared to conventional peptide screening methods, such as phage display, which are limited to a defined set of conditions and time-consuming, the method developed in this work is fast, effective, and offers flexibility and control over the experimental conditions. Moreover, this method can be used to investigate the effect of variables such as pH, solution speciation, temperature, and other variables of interest, on the adsorption performance of peptides. Overall, this screening method enables determining a set of key peptide sequences and conditions to achieve optimal adsorption and mineral selectivity. Furthermore, the method is of broad application and can be used to identify binding peptides for other minerals of interest and can be applied in other fields of research to study peptide binding onto other types of solid surfaces, such as determining protein or peptide adsorption on novel nanomaterial or developing biosensors.

Lastly, this study introduces the initial steps in utilizing peptide technologies for mineral processing by identifying promising peptides with affinities for target minerals, in this case, arsenic-bearing minerals. The ongoing investigation delves into understanding peptide–mineral interactions and aims to develop highly efficient, selective, and environmentally conscious flotation reagents.

**Author Contributions:** Conceptualization, S.B.e.A. and L.I.K.; methodology, S.B.e.A., L.I.K. and L.F.; software, L.I.K.; validation, S.B.e.A. and L.I.K.; formal analysis, L.I.K.; investigation, S.B.e.A. and L.I.K.; resources, S.B.e.A.; data curation, L.I.K.; writing—original draft preparation, L.I.K.; writing—review and editing, S.B.e.A. and L.F.; visualization, L.I.K.; supervision, S.B.e.A. and L.F.; project administration, S.B.e.A. and L.F.; funding acquisition, S.B.e.A. All authors have read and agreed to the published version of the manuscript.

**Funding:** The authors greatly acknowledge the financial support from Newmont Corporation to undertake this work. This work was supported by the ARC Centre of Excellence for Enabling Eco-Efficient Beneficiation of Minerals (grant number CE200100009) for personnel time of S.B.e.A. and L.F. in the preparation of this manuscript.

**Data Availability Statement:** Data are contained within the article. Due to privacy restrictions, additional data is not publicly available.

**Acknowledgments:** The authors would like to thank Chun-Xia Zhao from The University of Adelaide for the helpful discussions pertaining to the use of peptides.

**Conflicts of Interest:** The authors declare that they have no known competing financial interests or personal relationships that could have appeared to influence the work reported in this paper.

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
