# Peer review of "An Efficient Peptide Screening Method for Mineral-Binding Peptides"

_minerals, doi:10.3390/min14020207_

Round 1
Reviewer 1 Report
Comments and Suggestions for Authors
Dear Authors,
The paper presents a valuable research for the industry.
Please find below some suggestions that might help to improve the manuscript.
Lines 45-47:
To strengthen the paper, it might be beneficial to analyze research on bubble-particles attachment using thiol collectors with respect to metal sulphide minerals.
Please find below some examples:
Lines 73-91:
To sustain reading it might be beneficial to significantly reduce the last paragraphs of the Introduction section. Readers will appreciate if authors state the aim of the paper in a single sentence rather than describing the objectives. The objectives of the paper can be discussed in detail in the "Discussion" section.
Materials and Environmental Methods:
Readers will appreciate if authors describe environmental properties of peptides.
Line 99:
Please justify a reason for preparation of 20 μM solutions.
Lines 111-113:
The minerals were prepared using a resin, which can contaminate the surface of the mineral. This can affect the results of the adsorption tests. How did authors make sure the quality of the mineral surface in the adsorption tests? Did peptides adsorb onto clear adsorptive sites or onto resin-impurity ones?
3. Method Development:
Readers appreciate if authors specify the type of the peptide collector. Does cation, anion or molecule adsorb onto mineral surface?
Also, the paper might benefit if authors explain the reason for using the proposed adsorption method instead of widely known ones (IR, UV, etc.).
Lines 174-175:
I am afraid there is no "+" symbol in the last column of the Table 1. Suggest appropriate correction to make clarity for readers.
Please, check abbreviations in the first column (Capital letters?). They differ from the abbreviations in the Table 1.
Line 329:
Please check a formula for ferrihydrite.
5. Conclusions and Recommendations
The paper might benefit if authors shorten the conclusions. Please highlight key findings and their impact. Also, avoid any figures in the conclusions. The figures and summary of the method can be transferred to a separate subsection of the Discussion section.
In general, the paper is interesting to read and it reports novel results in mineral flotation.
Regards,
Reviewer
Reviewer 2 Report
Comments and Suggestions for Authors
This study explores the uncharted use of peptide technologies in mineral processing to develop novel flotation reagents. The goal is to create a method for screening mineral-binding peptides as potential reagent candidates, with a hypothesis that these peptides can selectively adhere to mineral surfaces. The research involves selecting peptide candidates from existing literature with an affinity for arsenic species and testing their adsorption on mineral surfaces under flotation conditions. The study successfully identifies target-specific peptides, demonstrating the method's efficacy and potential application in developing new reagent chemistries for various minerals in mineral processing.
The manuscript is very well-written, thought-provoking and has the potential to provide new directions to the field. However, I would urge the authors to provide a more extensive literature background in the manuscript. That will inform readers in a more comprehensive manner and will most likely lead to a more complete understanding and applications of their proposed technique.
Reviewer 3 Report
Comments and Suggestions for Authors
The manuscript An Effective Peptide Screening Method for Mineral-Binding Peptides deals with the possibilities of using peptides for the flotation of sulphide ores containing arsenic. This is an original idea, the research contains elements of novelty. The manuscript is written very systematically and clearly. The results and discussions are presented at a very good level. I have the following comments about the thesis:
1. The text on lines 201 to 204 is almost identical to the text on lines 218 to 221.
2. The thesis mentions the use of Image J software to quantify the adsorption of peptides on the mineral surface. Was this software also used to divide the amount adsorbed into categories: no, small, moderate, high? Or were the results presented in Tables 2, 3 and 4 obtained in a different way? For example, by estimation. It would be appropriate to add the exact procedure to the article.
3. The basic quantity that affects the adsorption of peptides on the mineral surface is the zeta potential. A large number of publications are devoted to the phenomenon of the zeta potential of sulfidic minerals. Zeta potential was probably not measured in this research, yet it would be appropriate to mention its importance in the article.
Notwithstanding all the above remarks, I recommend the manuscript for publication.
Reviewer 4 Report
Comments and Suggestions for Authors
Dear authors,
Your manuscript is very interesting from the aspect of the flotation process. I think the research direction is also good. However, I have some objections:
- The flotation conditions you specify, that is, the parameters you have chosen, should be clarified. Namely, state why you chose appropriate pH regulators and foaming agents? For what reasons, what are their characteristics? You can specify this in the description of the experiment.
- Here you do not provide any quantification of the binding efficiency of the selected peptides. On the basis of successful adsorption, you draw a conclusion on the selection of collectors for subsequent testing.?
- What is a small, and what is a large and moderate amount of fluorescence? This should be clarified.
- What do you think about increasing the hydrophobicity of minerals with your chosen peptide? I mean the peptide that shows no increase in mineral hydrophobicity after adsorption? Mention here in the discussion of the results the possibility of synergy with some of the xanthogens. This can be interesting. Especially if you do a complete flotation and show the mineral recovery and content in the tailings based on the results.
- Figure 1 should not be in the manuscript.
- Image 4. If you want it to remain, specify the text that indicates the image in front of the image. I would kick her out.
- At the end: Describe or present a quantification model of the results you reached in the part of the peptide binding to the mineral.
I wish you a happy New Year 2024
Best Regards
Round 2
Reviewer 1 Report
Comments and Suggestions for Authors
Dear Authors,
Although the paper has been improved, there are some typos that need correction:
The environmental impact means any activities of people and businesses have on the environment using the peptides reagents. Suggest correction at your choice and expertise.
The name of the section 4. "Results and Discussion and 4.8 "Discussion contains" the word "Discussion". Suggest remaining the "Results" and the "Discussion" sections.
Figure 4. The bottom of the text is cut off. The exponent is not fully displayed, etc. Please indicate the temperature conditions for the graph on the right side.
Suggest avoiding the citations in the Conclusion section.
Regards,
Reviewer
Author Response
Dear reviewer,
Thanks for pointing out the typos. They have now been fixed and the temperature for Figure 4 has also been added to the legend. The citations in the Conclusion section have also been removed.
Many thanks,
Lam
